# A Quality of Service-Aware Secured Communication Scheme for Internet of Things-Based Networks

**DOI:** 10.3390/s19194321

**Published:** 2019-10-06

**Authors:** Fazlullah Khan, Ateeq ur Rehman, Abid Yahya, Mian Ahmad Jan, Joseph Chuma, Zhiyuan Tan, Khalid Hussain

**Affiliations:** 1Department for Management of Science and Technology Development, Ton Duc Thang University, Ho Chi Minh City 71000, Vietnam; fazlullah@tdtu.edu.vn; 2Faculty of Information Technology, Ton Duc Thang University, Ho Chi Minh City 71000, Vietnam; 3Department of Computer Science, Abdul Wali Khan University Mardan, Khyber Pakhtunkhwa 23200, Pakistan; 4Department of Electrical, Computer and Telecommunication, Faculty of Engineering and Technology, Botswana International University of Science and Technology, Palapye 10071, Botswana; yahyabid@gmail.com (A.Y.); chumaj@biust.ac.bw (J.C.); 5School of Computing, Edinburgh Napier University, Edinburgh 00000, UK; Z.Tan@napier.ac.uk; 6Computer Science Department, Barani Institute of Sciences ARID University Sahiwal and Burewala, Punjab 57000, Pakistan; dr.khalid@baraniinstitute.edu.pk

**Keywords:** Internet of Things, security, sybil attack, Quality of Service, multi-hop flows, ad hoc networks

## Abstract

The Internet of Things (IoT) is an emerging technology that aims to enable the interconnection of a large number of smart devices and heterogeneous networks. Ad hoc networks play an important role in the designing of IoT-enabled platforms due to their efficient, flexible, low-cost and dynamic infrastructures. These networks utilize the available resources efficiently to maintain the Quality of Service (QoS) in a multi-hop communication. However, in a multi-hop communication, the relay nodes can be malicious, thus requiring a secured and reliable data transmission. In this paper, we propose a QoS-aware secured communication scheme for IoT-based networks (QoS-IoT). In QoS-IoT, a Sybil attack detection mechanism is used for the identification of Sybil nodes and their forged identities in multi-hop communication. After Sybil nodes detection, an optimal contention window (CW) is selected for QoS provisioning, that is, to achieve per-flow fairness and efficient utilization of the available bandwidth. In a multi-hop communication, the medium access control (MAC) layer protocols do not perform well in terms of fairness and throughput, especially when the nodes generate a large amount of data. It is because the MAC layer has no capability of providing QoS to prioritized or forwarding flows. We evaluate the performance of QoS-IoT in terms of Sybil attack detection, fairness, throughput and buffer utilization. The simulation results show that the proposed scheme outperforms the existing schemes and significantly enhances the performance of the network with a large volume of data. Moreover, the proposed scheme is resilient against Sybil attack.

## 1. Introduction

The latest developments in wireless technologies have allowed heterogeneous devices to form peer-to-peer networks. These devices, that is, smartphones, wireless sensors, smart visual tags and so forth, interoperate in a globally integrated communications platform. A set of self-organizing mobile and static devices communicate through wireless links to form a dynamic network. These multi-hop wireless ad hoc networks, that is, Mobile Ad hoc NETwork (MANET), Wireless Sensor Network (WSN), Vehicular Ad hoc NETwork (VANET), Radio Frequency IDentification (RFID) are considered as the backbone of the emerging Internet of Things (IoT). The IoT-based networks facilitate direct communication among the nodes (In this paper, we use the words nodes and devices interchangeably.) using off-the-shelf wireless standards such as Bluetooth, Infrared, WiFi, 4G and high-speed IEEE 802.11 protocol. Using these standards, the nodes communicate with each other via multi-hop wireless links. The intermediate nodes act as routers to forward data packets of other nodes for quality-of-service (QoS) provisioning. However, when the number of data transmitting nodes increases and generate a high amount of data, the performance of medium access control (MAC) layer (The Data Link Layer of the Open System Interconnection (OSI) model is divided into two layers, that is, the logical link control layer (LLC) and MAC layer. The job of LLC is flow control, error detection, error correction and framing, whereas the MAC layer MAC deals with accessing the channel.) protocols degrade dramatically in terms of fairness, throughput and delay [1]. This can further degrade QoS parameters such as loss of accurate information and delivery of information after due time. In dynamic IoT-based networks, every node uses the same medium, where at the MAC layer the allocated bandwidth for each node cannot guarantee per-flow fairness. Similarly, at the link layer, the direct flow (In ad hoc networks every node has to forward data in the network resulting in each node has to send two flows, *direct flow* and *forwarding flows*. A *direct flow* is a node‘s own flow and *forwarding flows* are the flows from other nodes.) and forwarding flows compete to access the buffer space and the direct flow has clear advantage over forwarding flows [2,3].

The Distributed Coordination Function (DCF) in the MAC protocol is intended to offer a fair opportunity to each node for transmitting its data [4]. The DCF uses carrier sense multiple access with collision avoidance mechanism and channel access using Binary Exponential Back-off (BEB) mechanism. The BEB is used to determine the contention window (CW) size according to network congestion by using Back-off Interval (BI) and CW. BI is decremented each time the channel is sensed idle and when BI = 0, a node starts transmission. The DCF provides an acceptable level of QoS provisioning for various ad hoc networks. However, in asymmetric multi-hop networks, BEB cannot fulfil the required level of QoS and provides low fairness and throughput, especially when a huge amount of data is generated by IoT nodes [5]. Moreover, BEB has no capability of providing QoS to forwarding flows, prioritization and on-time delivery of critical data. Note that, there is always a trade-off between throughput and fairness in a multi-hop wireless ad hoc networks [6]. The aforementioned challenges are due to the limitations of DCF because it cannot guarantee QoS due to the random access nature of BEB. The DCF supports only random access and it is unable to provide any service differentiation because all nodes have the same priority in accessing a channel. As a result, each node has the same CW${}_{min}$, CW${}_{max}$ and waiting time before back-off or retransmission, that is, Distributed Inter-Frame Space (DIFS). Various solutions have been proposed for achieving QoS in multi-hop flows using an enhanced CW size. For example, Cross-layer based on Utilization evaluation to Contention Window adaptation (CUCW) [7], Cooperation between channel Access control and TCP Rate Adaptation (CATRA) [8,9,10]. However, in these solutions, when a node accesses the channel, it transmits a higher number of packets without giving a fair chance to other nodes and the QoS issues are not considered.

The QoS provisioning is well-studied in ad hoc networks, where the main focus is to improve the QoS parameters. However, not much attention is given to the security requirements for QoS provisioning in ad hoc networks, which is an important issue. A secured QoS aware scheme is useful in many application, such as securing any critical network from Denial of Service and fabrication attacks. The integrity of data and on-time delivery of information are the main requirements of patient monitoring and surveillance systems. Moreover, in these systems the QoS is another important factor that provides low delay and high throughput of the network by providing a fair chance to nodes for using the communication channels. In References [9,10,11,12], we have studied the performance improvement in wireless ad hoc networks. However, we also did not consider the security requirements or the possible attacks on a network that may degrade the QoS. The IoT-based dynamic networks are made on the run and operate in a wireless environment. The adversaries can easily capture and maliciously manipulate any information exchange via wireless channels. Therefore, an adversary may transmit data with multiple identities at the same time, that is, Sybil attack. A Sybil attack can cause Denial of Service (DoS), impersonation and other attacks. Thus degrading the QoS provisioning, apart from the Link and MAC layer issues related to ad hoc networks.

In this paper, we proposed a QoS-aware secured communication scheme for IoT-based networks (QoS-IoT). Our main contributions to the literature can be summarized as follows:We propose two algorithms; the first, a lightweight protocol for Sybil nodes detection, that is, a signalprint-based (A signalprint is created from the received signal strength information of a node to detect misbehaving nodes.) Sybil attack detection. This protocol has devised two policies for detection of Sybil nodes. The Sybil nodes are detected by high-power and mobile nodes are reported to the genuine nodes in the IoT-based network; as a result, genuine nodes do not entertain Sybil nodes.The second proposed algorithm is an adaptive algorithm for determining the optimal size of the CW and allocates the bandwidth to the nodes based on the current network status. This algorithm helps in maintaining a balance between per-flow fairness and fair allocation of bandwidth.For the QoS provisioning, the proposed QoS-IoT scheme uses a mechanism where CW size is determined based on the ratio of actual to fair bandwidth allocation. Different CW size is assigned to different flows for fairness, that is, smaller CW size to is assigned to flows having more substantial queue length.Finally, we perform extensive simulations to prove the efficacy of the QoS-IoT in terms of fairness, throughput and link utilization. The simulation results are compared with the existing schemes.

The rest of the paper in accordance with the following pattern. In Section 2, related work is presented, followed by the system model in Section 3. The fairness problems in multi-hop ad hoc networks are discussed in Section 4. Section 5 describes the proposed scheme and experimental work and evaluations of the proposed scheme are provided in Section 6. The paper is concluded and future research directions and gaps are discussed in Section 7.

## 2. Related Work

The Internet of Things (IoT) is mainly based on several matured and related technologies, that is, MANETs, WSN, RFID devices and so forth Specifically, WSN is the networks of things; MANET is the network of people; RFID is the network of car’s parking slots. These networks are dynamically created and allow things and people in a restricted area to exchange data without any infrastructure. As a result, IoT-based ad hoc networks got great attention in the research community. The wireless standard for ad hoc networks, that is, IEEE 802.11, defines two operational modes, infrastructure-based and infrastructure-less or ad hoc mode. The IEEE 802.11 is a good platform to implement single-hop ad hoc networks due to its low cost and efficiency in avoiding collisions with simple mechanisms. This limitation can be overcome by multi-hop ad hoc networking [1,5]. The ad hoc wireless network is created when nodes intend to communicate with each other or with a group. Every node in the network is willing to forward data packets of other nodes when they are not in transmission range. The multi-hop ad hoc networking concept is used in various applications like virtual classrooms and conference rooms in academia, MANETs, VANETs, WSNs and IoT. In ad hoc wireless networks, the purpose of the MAC protocol is to use scarce resources efficiently. The efficient use of bandwidth facilitates the network to fulfil application-specific requirements like fairness, energy consumption, QoS, throughput and robustness.

For achieving QoS in Ad hoc networks, a cross-layer scheme was investigated for controlling CW in asymmetric multi-hop networks [7,8]. Due to cross-layer signalling, this scheme performs well in terms of throughput and fairness. However, in this scheme, a node accessing the medium first will transmit all its packets and then will give a chance to other nodes. As a result, the transmission time for direct flow is almost double than the forwarding flows [13]. Moreover, link utilization is not good in long-chain topologies [11], circular topologies and does not perform well on mobile topologies [10]. In Reference [14], the authors proposed an analytical model that works only when the network is saturated. They studied the increase of queue length at the link layer and highlighted their correlation, that is, overflow flow and empty queues at the same time. The proposed model performs the load balancing among the nodes with limited buffer space. However, none of these papers has considered security requirements for QoS provisioning. Similarly, in Reference [15], the authors have proposed FogTorch, a QoS-aware IoT infrastructure. This infrastructure helps in the deployment of critical functionalities on large-scale and heterogeneous IoT networks.

Like other communication networks, IoT-based networks use wireless channels. An adversary can easily inject malicious data to a communication channel, especially when multi-hop communication is intended [16,17]. In such situations, Sybil nodes can degrade network performance by forging multiple illicit identities at a particular time [18]. A Sybil node uses multiple illicit identities by forging legitimate nodes [17]. Various Sybil attack models are studied in the literature; the possible threats to IoT devices are list as follows. In Reference [19], the authors have proposed a Sybil attack detection mechanism based on the count interval and affinity value of the observer and Sybil node. The affinity value was calculated using a graph. In Reference [20], the authors have used a watchdog with a unique label to identify the mobile Sybil nodes in the network. The detection of a Sybil node is challenging based on the probability of two nodes having the same neighbours in a densely deployed network [21]. Similarly, in Reference [22], the authors have proposed a Sybil attack detection mechanism based on the principle that the RSS of the first legitimate node is low when it enters the radio range of a receiver. In Reference [23], an analytical model of Sybil attack detection in the IoT environment is provided. This model works on three phases, that is, compromise phase, deployment phase and launching phase. In the first phase, the attacker is detected using a Markov chain model. In the second phase, a k-mean clustering approach is used to identify compromised identities. In the last phase, the Sybil identities are detected and replaced. In Reference [24], the authors have studied the effects of Sybil nodes in the network performance. The Sybil nodes advertise fake optimal paths using their illicit identities. In Reference [25], the authors have studied the effects of Sybil nodes on the result of voting or clustering head selection. A Sybil node can select or reject a legitimate node by using multiple, forged identities. In Reference [26], the authors have studied the effects of Sybil nodes on data aggregation in the IoT environment. The Sybil nodes may take part in data collection and machinate the collected data by giving false-negative reports. Furthermore, the nodes may report wrong time-stamps using illicit identities and forged the data. In References [16,17] the authors have studied the effects of Sybil nodes on user’s privacy. The Sybil node becomes a member of the IoT cloud storage nodes using forged identities. The Sybil nodes then allow the attackers into the cloud storage and cause privacy breaches. Like the existing schemes, the main limitations of our proposed scheme is that it may not work efficiently in a very dense network and movable networks like flying ad hoc networks, Internet of Vehicles and Internet of drones. The possible reasons are the dynamic topology and ad hoc nature of data transmission with various signal strengths. Furthermore, we have not considered the energy consumption of nodes for Sybil attack detection. As the Sybil node detection consumes a considerable amount of the node’s energy.

## 3. System Model

In this section, we discuss the system model of the proposed scheme is shown in Figure 1, where different IoT-based networks cover a city of 100 × 100 Km${}^{2}$. The total area is divided into smaller IoT-based networks, where each network consists of Sybil, mobile, static and high power nodes.

The mobile nodes move at an average speed of 10 Km/h and change their positions within a network after every 100 s. The high power nodes directly transmit data to the base station, whereas mobile nodes forward data of static nodes along with their own data to the Base Stations. Moreover, the mobile nodes help in the detection of Sybil nodes and do not forward their data to the Base Stations. The Base Stations further transmit the received data to the Data Centers via the Internet (This concept is out of the scope of this paper.).

In recent years, the deployment of mobile nodes in IoT-based networks gained popularity in the research community. However, in multi-hop mobile communication, all nodes have equal priority and that is why the QoS issues are not considered. This claim can be justified from the simulation results of a basic multi-hop topology, depicted in Figure 2. For example, in Figure 2a, when a node captures the medium, it transmits a higher proportion of its packets, while in the proposed scheme, each node gets a fair chance to access the medium as depicted in Figure 2b. In a basic multi-hop (two-hop) topology, Node${}_{1}$ has to forward flows of Node${}_{2}$ to the receiver with fairness. However, in the original MAC protocol, Node${}_{1}$ does not give a fair chance to Node2, as shown in Figure 2a. The results depicted in Figure 2a show that Node${}_{1}$ (blue color) has high throughput; this is because it transmits huge number of packets compare to Node${}_{2}$ (orange color). In contracts, the results depicted in Figure 2b, both nodes have a fair chance of transmitting their packets.

From the QoS perspective, we mainly consider delay, throughput, because packet loss and jitter can be enhanced using retransmission of packets and their buffering, respectively. This is why we have also considered buffer utilization in our proposed scheme. One of the reasons to study QoS is that IoT-based networks involve a large number of heterogeneous devices. These devices generate, collect, process and transmit huge and complex data with different QoS requirements. The QoS provisioning and Sybil attack detection are the themes of this paper. The QoS metrics like delay and throughput are profoundly affected by the presence of Sybil nodes. During communication, the Sybil nodes prevent legitimate nodes from communication by occupying network resources with different forged identities. The purpose of Sybil nodes is to send false-negative alerts and poses various threat to critical data. To achieve the QoS in the proposed scheme, we devised an algorithm for Sybil nodes detection along with optimal CW selection. The optimal CW is selected based on the BEB mechanism. In BEB, a BI value is selected based on a uniform random distribution as given in Equation (Equation 1)
(1)$$BI=Unif.Rand(0,CW_{i}).$$

The CW size is selected based on Equation (Equation 2)
(2)$$CW_{i}=\left\{\begin{array}{ll} CW_{min} & \mathrm{for}\;i=0,\\ min(2*\left(CW_{i-1}+1\right)-1,CW_{max}) & \mathrm{for}\;i\;\neq \;0 \end{array}\right.$$
where, *i* is the number of consecutive failed attempts due to collision or busy medium.

## 4. Quality of Service Degradation

This section describes the reasons and issues that cause poor performance in multi-hop wireless ad hoc networks. These issues are categorized into MAC layer problems, Link layer problems and malicious node activities.

### 4.1. Problems at the MAC Layer

Wireless MAC protocols are classified into distributed MAC and centralized MAC protocols. The design of these protocols is very challenging compared to wired protocols due to half-duplex operation, time-varying channel and burst-channel errors. Apart from this, MAC layer suffers from well-known issues, such as the hidden terminal problem [27], expose terminal problem [28], capture terminal problem [29], extended inter-frame space problem [30] and the three-pair scenario problem [6].

### 4.2. Problems at the Link Layer

The direct flow and forwarding flow contend in the outgoing buffer space at the link layer. The detail descriptions of the problems in first-in-first-out (FIFO) and round-robin (RR) scheduling algorithms can be studied from our previous work [9,10].

### 4.3. Malicious Node and Trust Model

Due to the decentralized nature of IoT-based ad hoc networks, trust becomes a critical issue in such networks. It is a challenging task to determine whether the next-hop neighbour, that is, a relay, in a transmission path, is a trustworthy node or a malicious one. Furthermore, it is also important to determine if the transmitter is a genuine node or a Sybil node. The Sybil attacks can be of two types, that is, single identity-based attack and multiple identities-based attacks. In the former type, the malicious node uses only one fake identity at a time and the purpose behind this attack is to clean-out the history of any previous attack. In the latter attack, one malicious node utilizes multiple identities simultaneously and the motive is to gain more network resources to bring down the network performance. In this paper, we consider both types of attacks and we aim to design a strategy to detect either type of identities, created by a malicious node.

## 5. QoS-Aware Secured Communication Scheme

In this section, we will explain our proposed QoS-IoT as a secured communication scheme and prove its suitability for ad hoc networks. The main focus is on the QoS provisioning and the detection of Sybil attack.

### 5.1. Security Attack Detection and Prevention Model

The proposed model works on the signalprints-based Sybil attack detection. The proposed model does not require any prior knowledge about the network deployment and is able to determine if the next-hop node is an adversary or a genuine node. We consider the signalprint as a vector of RSSI from multiple transmitters. This vector V is a combination of transmission power P and attenuation A, as shown in Equation (Equation 3).
(3)V=P+A
where V can also be considered as a function of the receiver’s characteristics and amplitude response of the transmission channel.

A group of nodes can easily be classified as Sybil nodes if both the observing nodes and the initiator node reports the same RSSI for a specific node or a group of nodes. The initiator is a node that trusts its own RSSI and does not trust anyone else. It has the ability to label a node as either Sybil or non-Sybil. Each non-Sybil node is an initiator node and becomes an observer when it shares its knowledge with its neighbours. Sybil nodes try to adopt different tricks to appear as genuine nodes. The purpose of this model is to identify and report a node as either True (genuine) or False (Sybil), based on their RSSI observations without any previous knowledge about the participating nodes. Let S denotes a set of participating nodes in the network that contains both Genuine Nodes (GNs) and Sybil Nodes (SNs). After classification, two subset are created, where,
(4)$$\begin{array}{c} GN\in S_{1},\;\;\;S_{1}\subset S\\ SN\in S_{2},\;\;\;S_{2}\subset S. \end{array}$$

Each participating node maintains a classification knowledge (*K*) about GN and SN and shares it with its neighbours. If the neighbours agree with the generated *K*, it is known as knowledge with true classifications and is denoted by $K_{t}$. On the other hand, if anyone among the neighbouring nodes objects to the generated *K*, it is labelled as knowledge with false information and is denoted by $K_{f}$. In most of the cases, no method can produce 100% accurate *K*. In this model, two different security policies are combined to identify Sybil nodes. Each policy follows different rules for nodes classification.

### 5.2. Sybil Node Detection

In this section, we discuss nodes classification policies. In the first policy, every node in the network generates a report $K_{i}$. The $K_{i}$ generated by all the nodes is compared with each other. Whichever $K_{i}$ reports the maximum number of Sybil nodes, is considered as the most authentic knowledge ${\bar{K}}_{i}$. Secondly, ${\bar{K}}_{i}$ is shared with all the nodes except the one who generated it by turning their knowledge into ${\hat{K}}_{i}$. This policy generates knowledge report with running error and creates two situations, that is, (1) the maximum number of SN contain GN too and (2) it is possible that there exists more SN which are not included in the generated knowledge report; thus, the report’s threshold for the maximum number of SN cannot be considered authentic.

The second policy is based on two conditions, that is, (1) in all generated reports, each report must contain GN and (2) the GN must be higher in number than SN. These conditions can easily be met. The GN remain consistent in their observations, while conflicts can be found in SN observations due to their random identities. A set of true nodes (TN) can be considered as True if and only if it is present in all the reports generated by other nodes and truly classify GN and SN, that is,
(5)$$\begin{array}{c} TN\subset S\\ TN\cap S_{1}\neq ⌀\\ TN\cap S_{2}\neq ⌀ \end{array}$$

A report can either be fully correct or partially correct. If utilized properly, report consistency can be helpful to increase the number of GN in each report. A malicious node may generate a consistent report, claiming some SN as GN and vice verse. This report is partially true but can collapse other true reports, generated by other nodes. However, such reports still classify at least one node as GN. It is also possible that a report generated by a node is 100% true report. Such a report can be considered as a base in turning the partially true reports to wholly true reports. However, this policy is based on assumptions and circumstances and cannot be applied in practice. The Sybil node detection based on the above policies is summarized in Algorithm 1.

### 5.3. QoS Aware Communication

In QoS-IoT, we have considered contention-based medium access mechanism of the IEEE 802.11 protocol, that is, the DCF. In multi-hop networks, the DCF does not distinguish flows and that is why direct flow causes QoS violations for forwarding flows. The DCF grants channel access to each flow using its CW size. As a result, the forwarding flows get lesser chances to access the medium due to contention at MAC and link layers. Similar to our previous work [9,10,11,12], in QoS-IoT, we address QoS issues at MAC and link-layer using cross-layer signaling. The link-layer contention is the main reason that the direct flow occupies the buffer completely during heavy load. Thus, the unfairness problem cannot be solved by using round-robin queues only. At the link layer, the QoS-IoT compares the queue length of each flow with the average length of all flows and mark packets to give a fair chance to packets from forwarding flows. The packets are delayed which are marked using Equation (Equation 6).
(6)$$Marked=\left\{\begin{array}{cc} 0 & \mathrm{if}\;\ell_{i}\leq \sigma \\ 1-\frac{\ell_{i}\;-\;\sigma }{(n+1)\times \sigma }, & \mathrm{if}\;\ell_{i}>\sigma \end{array}\right.$$
where, $\ell_{i}$ is the queue length of flow *i*, σ is the average queue length of all flows and *n* is the number of flows. The QoS-IoT uses a cross-layer signalling to adjust CW size based on each flow’s information collected from physical, MAC and link layer and select a new CW size. This newly selected optimal CW size gives fair chances to forwarding flows for accessing the channel. The optimal CW selection is computed using Algorithm 2.

**Algorithm 1** Sybil Node Detection**Initialization: K** = ⌀, i = 0, α = 1, β = 1, γ = 1, δ = 1.
**procedure**

2:
    K ← K${}_{t_{i}}$
4:
    **while**
i<η**do**▹η is the number of nodes.
6:
       **if** K${}_{t_{i}}\geq$ K${}_{t_{i+1}}$
**then**
8:
        K${}_{t_{i}}$ = K${}_{t_{i}}$
10:
        *i = i* + 1
12:
       **else**
14:
         K${}_{t_{i}}$ = K${}_{t_{i+1}}$
16:
       **end if**
18:
    **end while**
20:
    **return** K${}_{t}$
22:
    **broadcast** K${}_{t}$ to all nodes in the network.▹ After this broadcast, Node 0 will continue the following policy
24:
    (K, R${}_{max}$) ←(∞,⌀)
26:
    **while** R${}_{\alpha }\in$ K **do**
28:
       **if** R${}_{\alpha }$(SN${}_{\beta }$) ≠ RSSI(R${}_{\alpha }$(SN${}_{\beta }$)) **then**
30:
        Exclude R${}_{\alpha }$(SN${}_{\beta }$)
32:
       **else**
34:
         RR${}_{\gamma }$← R${}_{\alpha }$(SN${}_{\beta }$)
36:
       **end if**
38:
       **if** R${}_{\alpha }$(GN${}_{\beta }$) ≠ RSSI(R${}_{\alpha }$(GN${}_{\beta }$)) **then**
40:
        Exclude R${}_{\alpha }$(GN${}_{\beta }$)
42:
        α = α + 1
44:
       **else**
46:
         RR${}_{\gamma }$← R${}_{\alpha }$(GN${}_{\beta }$)
48:
         γ = γ + 1
50:
       **end if**
52:
    **end while**
54:
    γ = 1
56:
    bool = true
58:
    **while** bool **do**
60:
       **if** SizeOf(RR${}_{\gamma }$(GN)) ≥ SizeOf(RR${}_{\gamma +1}$(GN) **then**
62:
         R${}_{max}$ = RR${}_{\gamma }$(GN)
64:
         γ = γ + 1
66:
       **else**
68:
         R${}_{max}$ = RR${}_{\gamma +1}$(GN)
70:
       bool = false
72:
       **end if**
74:
    **end while**
76:
    **return** R${}_{max}$
78:
    **broadcast** R${}_{max}$ to all nodes in S.
80:

**end procedure**


**Algorithm 2** Optimal Contention Window Selction**Initialization:** 𝚥 = 0, 𝚤 = 0, T${}_{x}$ = 0.
**procedure**

2:
    count 𝚤▹𝚤 is a flow sensed at the physical layer by a node, which is out of transmission range but within the sensing range
4:
    count 𝚥▹ 𝚥 is a flow within the transmission range
6:
    **for** each TS duration **do**▹ MAC layer is divided into Time Slots (TS) of fixed intervals
8:
    𝚥’s duration = 10% × 𝚥’s time + 90% × T${}_{x}$ duration▹ time taken by 𝚥 flow
10:
       $T_{x}$ = 0
12:
       **for** each packet *p*
**do**
14:
          **if**
*p* is *CTS*
**then**
16:
            T${}_{x}$ = T${}_{rtc}$ + T${}_{cts}$
18:
          **elseif**
*p* is *ACK*
**then**
20:
            T${}_{x}$ = T${}_{data}$ + T${}_{ack}$
22:
           end **if**
24:
        end **for**
26:
    end **for**
28:
    $R=\frac{𝚥}{TS\;duration}$▹ R is real allocation of bandwidth
30:
    F = $\frac{𝚥}{𝚤+𝚥}$▹ F is the fair allocation of bandwidth to a node
32:
    CW${}_{adjusted}$ = CW $\times \frac{R}{F}$
34:
    CW${}_{optimal}$ = $\kappa \times CW_{adjusted}$▹ if packet is marked using Equation (Equation 6) and κ > 1 is a delay factor.
36:
    CW${}_{optimal}$ = CW${}_{adjusted}$▹ if packet is not marked using Equation (Equation 6)
38:

**end procedure**


## 6. Results and Discussion

To evaluate our proposed system, we performed a simulation-based investigation for encountering the effect of Sybil nodes on various QoS attributes. In this paper, a scenario with node mobility in the IoT network was considered. The simulations conditions are shown in Table 1. Further, the results elaborate that the optimum values of CW and the Sybil nodes detection have a significant impact on the performance of QoS-IoT. In this section, we will discuss the performance results in terms of fairness, throughput and link utilization against the FIFO, RR scheduling and CUCW using NS-2 [31].

### 6.1. Fairness Index

Fairness index is defined by R. Jain [32] as given in Equation (Equation 7),
(7)$$Fairness\;Index=\frac{{\left(\sum_{i=0}^{n}x_{i}\right)}^{2}}{n*\sum_{i=0}^{n}x_{i}^{2}},$$
where *n* represents number of flows, $x_{i}$ is the throughput of $flow_{i}$. The upper bound of fairness index is 1 and its lower bound is 1/n. In the worst case, the fairness index approaches the lower bound and in the best case, it reaches the upper bound. The fairness index against different offered load FIFO, RR, CUCW and QoS-IoT, are illustrated in Figure 3.

It is justified from the figure that due to the optimal selection of CW size, the fairness index of our proposed scheme QoS-IoT is higher than FIFO, RR and CUCW. One of the reasons for higher fairness index is that the proposed scheme gives a fair chance to access channels by forwarding flows. Moreover, when the offered load is high, each the fairness index changes every time due to contention between the direct flow and forwarding flows as well as contention at the MAC layer, that is, large-EIFS & 3-pair problems. This is due to the contention at the link and MAC layer resulting in very low fairness index of FIFO. The offered load get higher when sensor nodes in the IoT-based network generate various types of traffic and forward it to high power nodes or mobile nodes. These nodes receive a huge amount of data and then forward this data to the base station. In other words, the high power and mobile nodes are burdened with data of the network when it is transmitted through multi-hop links. In this scenario, the farthest nodes do not get a fair chance of transmitting data to the base station. Like FIFO, the RR algorithm cannot guarantee QoS because it works at the link layer and is unaware of the flows out of the transmission range. Hence RR cannot solve unfairness at the MAC layer. On the other hand, the proposed scheme achieves the best per-flow fairness, because, it improves the fairness at both MAC and link layers. In the proposed scheme when the offered load is large, $CW_{adjusted}$ is adjusted to a large value for directed flows and packet marking probability is higher which increases $CW_{optimal}$ for direct flows. The larger $CW_{adjusted}$ size decreases the chances of accessing the channel for direct flows at the MAC layer, whereas the larger $CW_{optimal}$ size decreases the chances of accessing the queue for direct flows. Like Figure 4 and Figure 5 also shows a comparison of the fairness index for multi-hop flows. In Figure 6, the effect of Sybil nodes is clearly shown. The Sybil nodes send many packets and do not give a fair chance to other nodes due to which the fairness index is minimum.

### 6.2. Throughput

The throughput is defined as the number of bits transferred in a particular time interval. The optimal CW size also provides better throughput compared to RR and CUCW, as shown in Figure 4. The highest throughput of FIFO is due to the transmission of a larger number of direct flow packets continuously. Whereas, the proposed scheme gives better throughput due to a fair chance to forwarding flows; as a result, the throughput of all forwarding flows increase, which causes an increase in total throughput. In the proposed scheme, the increase in the total throughput is because of the smaller $CW_{adjusted}$ and $CW_{optimal}$ sizes for forwarding flows due to which many packers are sent. In this way, the throughput of forwarding flows increased while the throughput of direct flows slightly decreased.

When the offered load is high, any node that access the channel consumes the whole bandwidth in FIFO, resulting in high total throughput but very low fairness. The CUCW method solves this issue but not as good as proposed scheme because in CUCW method an advantaged node is always in advantage and forwarded flows do not get many chances of transmission, as depicted in Figure 2. In the proposed scheme, the MAC layer fairness is ensured so that every node get its share in the channel bandwidth. In the proposed scheme, all nodes access the channel equally and send more packets than the CUCW method but fewer packets compare to FIFO due to their unfair nature.

Similarly, in Figure 5 the effect of Sybil nodes (QoS-Sybil) on total throughout is shown. In this figure, the total throughput of the QoS-Sybil is minimum because the Sybil nodes send false data and waste the network resources, that is, bandwidth. In this way, the actual nodes do not get fair chances of accessing the channels.

### 6.3. Average Queue Length

Average queue length is the average of all queues in a node during the simulation. The queue length *ℓ* of the direct and forwarding flows is shown in Figure 7. When the offered load smaller, that is, the sum of offered loads from all flows is smaller than the available bandwidth, resulting in a smaller *ℓ* for all the scheduling methods. Whereas in the case of high offered load, the shared queue in FIFO and the direct flows queue in RR schedule is full of packets.

The CUCW method faces same problem as RR because CUCW method is based on RR scheduling. There are many packets in the queue using FIFO and RR scheduling; this is due to unfairness at the MAC layer and link layer. However, in the proposed scheme, all the nodes get fair access to the channel and send many packets in time, that is why *ℓ* in the proposed scheme is better than other schemes. Similarly, in Figure 8, the effect of Sybil nodes on queue length is depicted. The largest queue length of QoS-Sybil is due to the Sybil nodes in the network. The Sybil nodes send malicious data and use network resources to degrade their performance. As a result, the actual nodes can not get fair chances of accessing the channels and the queue length increases.

## 7. Conclusions

In this paper, a QoS-aware secured communication scheme for IoT-based networks (QoS-IoT) is investigated in terms of identifying and preventing Sybil nodes and their attacks. The effects are elaborated in simulations. Moreover, the performance of scheduling algorithms are quantified and compared; for example, first-in-first-out (FIFO) and round-robin (RR) scheduling give poor performance. This is because, the FIFO scheduling shares a single queue among all flows, due to which it cannot solve queue contention. The RR scheduling is not efficient as contention at the MAC layer does not allow enough bandwidth for forwarding flows. On the other hand, the CUCW algorithm has a lower throughput. When a node acquires a channel, it occupies it till the transmission of all its packets. As a result, other nodes in the network are unable to access the channel that causes lower throughput and fairness. In contrast, the proposed QoS-IoT works on cross-layer signalling and achieves better results in fairness and total throughput and enhances the performance using Sybil nodes detection. The efficiency of the proposed scheme has been justified via the simulation results. These results show that our scheme outperforms the existing schemes and significantly improves the performance of overloaded networks. In future, we plan to study the effect of Sybil node detection with QoS on the Internet of Vehicles and flying ad hoc networks.

## Figures and Tables

**Figure 1 sensors-19-04321-f001:**
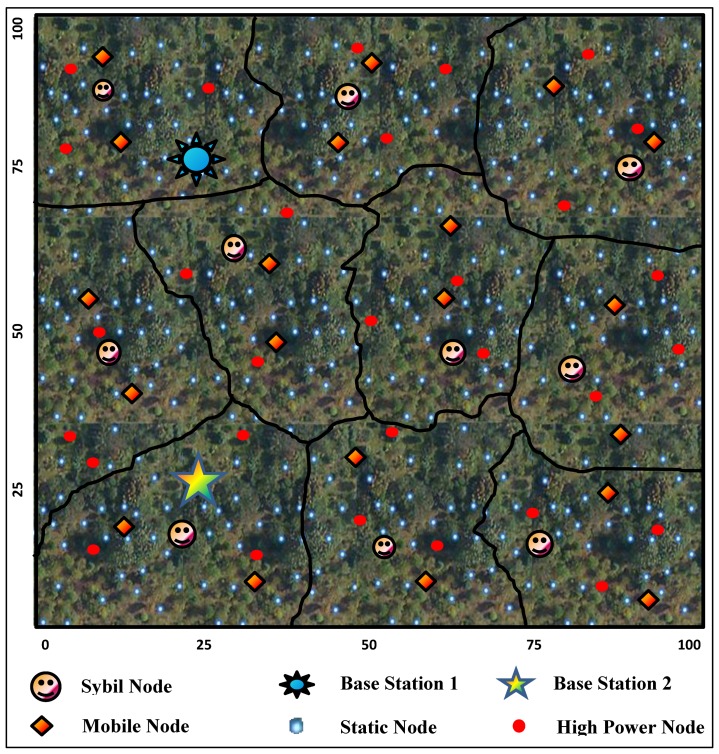
System Model of the Proposed Scheme.

**Figure 2 sensors-19-04321-f002:**
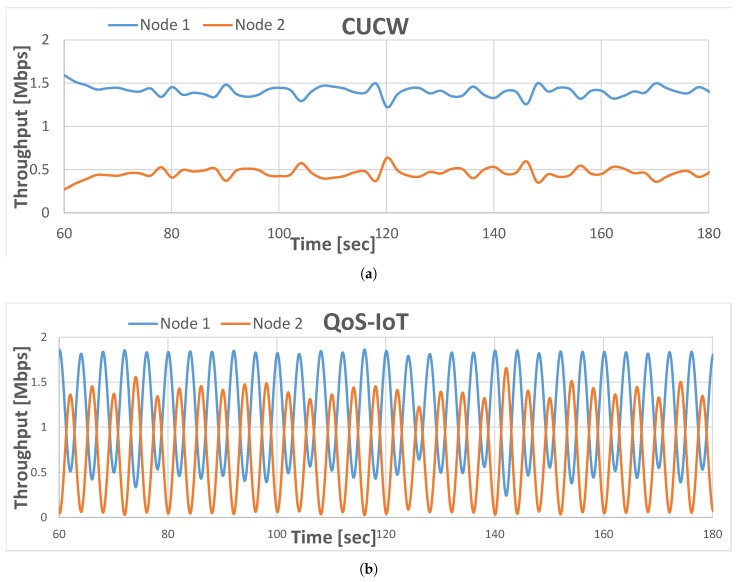
Simulation Results of CUCW [7] (**a**) and QoS-IoT (**b**).

**Figure 3 sensors-19-04321-f003:**
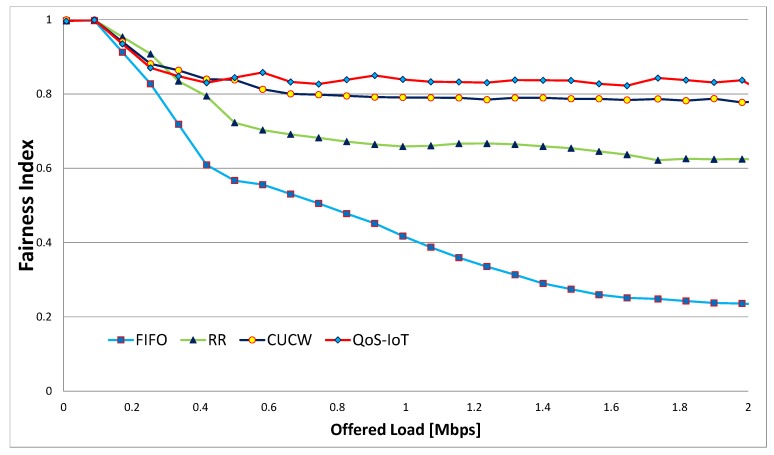
Fairness Indices with Sybil Nodes detection.

**Figure 4 sensors-19-04321-f004:**
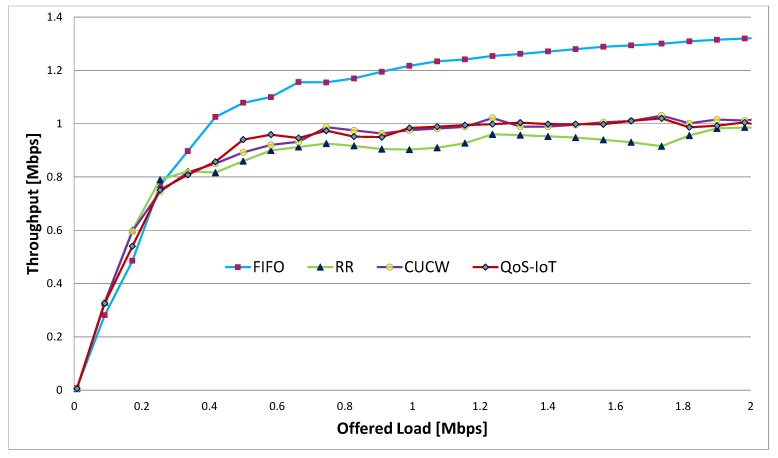
Total throughput for various schemes.

**Figure 5 sensors-19-04321-f005:**
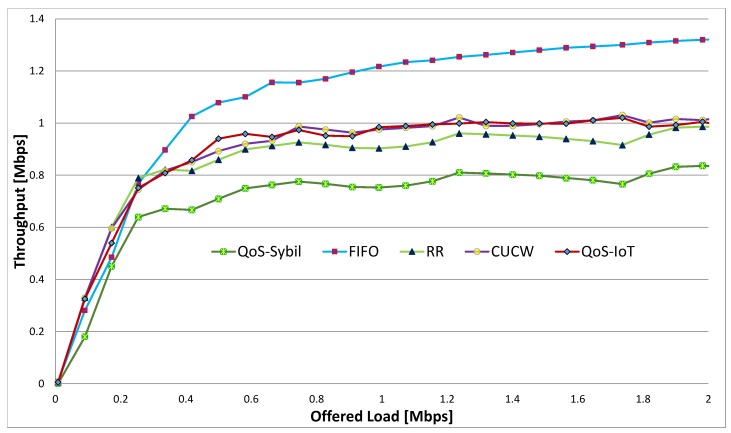
Total throughput and the affect of Sybil Nodes.

**Figure 6 sensors-19-04321-f006:**
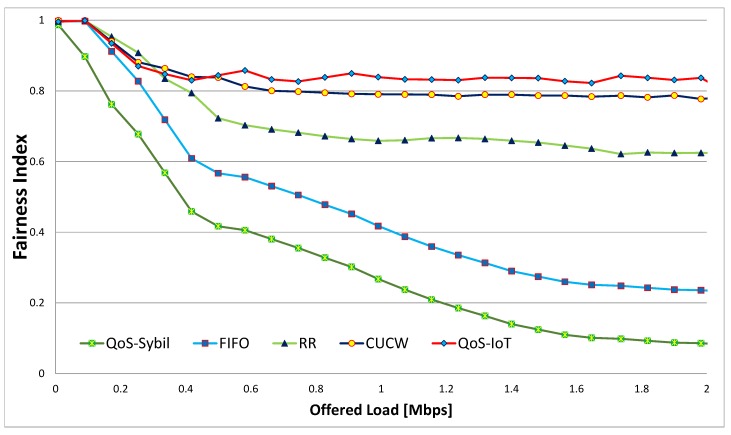
Fairness Indices affected by Sybil Nodes.

**Figure 7 sensors-19-04321-f007:**
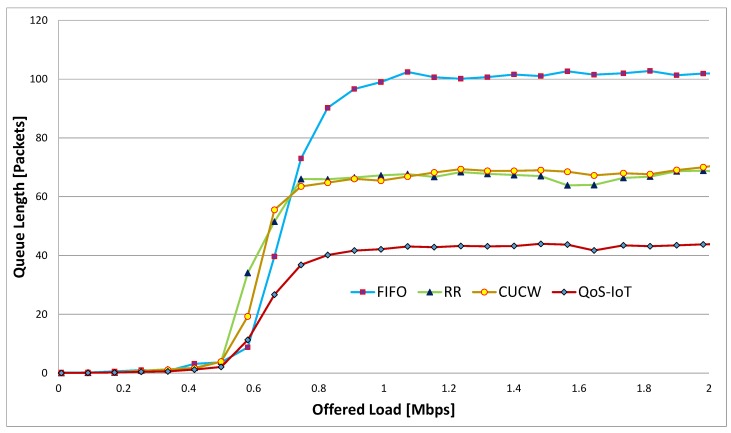
Average queue length in of all flows.

**Figure 8 sensors-19-04321-f008:**
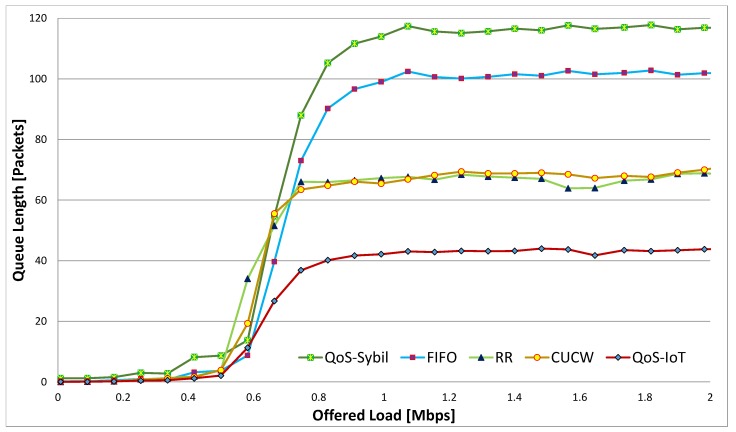
Average queue length in of all flows in the presence of Sybil Nodes.

**Table 1 sensors-19-04321-t001:** The simulation conditions.

Parameters	Values
Channel data rate	2 [Mbps]
Antenna type	Omni direction
Radio Propagation	Two-ray ground
Transmission range	450 [m]
MAC protocol	IEEE802.11b
Routing protocol	AODV
Connection type	UDP/CBR
Maximum Queue length	100 [packet]
Distance between stations	300 [m]
Number of nodes	random
Packet size	1024 [Byte]
κ	2
Simulation time	1000 [s]

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
