# Peer review of "A Quality of Service-Aware Secured Communication Scheme for Internet of Things-Based Networks"

_sensors, 2019, doi:10.3390/s19194321_

Round 1
Reviewer 1 Report
The paper is fairly written and properly organized. It seems that it has a merit, however some issues ought to be resolved.
Major issues:
1. Please clearly indicate the importance (significance) and area of implementation (application, usage) of the proposed algorithms.
2. Please clearly point out limitations and weak points of the proposed solution.
3. Please improve the “Conclusions” section. It does not refer to the proposed algorithms, nor shows future research directions.
Minor issues:
1. Please expand all abbreviations (e.g. “MAC”).
2. Please check carefully manuscript in regards of typos and language mistakes (e.g. “Selction”).
3. “Algorithm 2” is not properly formatted (see lines 36-38).
Reviewer 2 Report
The paper deals with a really interesting topic (QoS communication scheme
for IoT-based networks with detection of Sybil attack nodes) and it
adopts a very practical approach.
I have some insights and comments. To help the reader and to improve
the quality of the manuscript I suggest to modify/consider the
following aspects:
l.35-36: The terms "MAC layer" and "data link layer" can be equated by
the reader. For this reason, it worth defining this terms on paper and
providing what are relationship between these layers.
l. 42, 223: As a general rule, before using a abbreviation, you must define its full name. In most cases this rule is applied, but there are exceptions, e.g. l.42 "BI" and l.223 "RR". These abbreviations are defined in the glossary at the end of the paper. Maybe it's worth mentioning the existence of a glossary at the end of the paper in the footnote to the first shortcut used.
The paper has unnumbered lines between the numbers: 152-153, 184-185, 186-187, 202-203 and 211-212.
l.152: The first paragraph after line 152: The description of Figure 2 is very unclear. There is no connection between this description and the content of Figure 2. It is not described how the node Node1 differs from the Node2 node.
Section 5.2: There is a lack of estimation or at least information on how the described approach affects the computational load for each node and how the network is burdened with data exchange for these calculations.
Algorithm 1 (1): The purpose of the variables is not defined: i, epsilon, rho, gamma, and delta. Their definition will allow an easier understanding of the algorithm.
Algorithm 1 (2): Throughout the algorithm, constructs type i = i + 1 appearing in both branches of the "if" statement should be moved after the "if" statement.
Algorithm 1 (3): The "while" loop starting on line 60 has no condition for terminating the loop - this loop is infinite.
Formula (7): Formula is incorrect - the left parenthesis is missing.
Minor editing errors:
l. 69: there should be a ":" at the end
l.70: before the word "first" there should be "the"
l.81: should be "efficiency"
l.91: RFID is not a car network, but VANET - yes
l.115: reference number is missing in "[]"
l.123: there should be a period after the word "follows"
l.149: after the words "high power" there should be the word "node"
l.242: there should be the word "multi-hop"
Round 2
Reviewer 1 Report
Dear Authors,
All my comments have been properly addressed, thank you. Good work!
Author Response
Dear Reviewer,
Thank you very much for appreciation and motivation.
Reviewer 2 Report
I would like to thank the authors for their effort in correcting the paper and preparing an answer to the reviewer's comments.
I have two more observations:
l. 162. If the word "nodes" was used, then the word "transmit" should be used later in the sentence, not "transmits".
Algorithm 1, line 46: The index "b" is not defined in Algorithm 1. It is probably a beta index.
Author Response
Dear Reviewer,
Thank you very much for the insightful comments. We have made the necessary corrections as follow.
l. 162. If the word "nodes" was used, then the word "transmit" should be used later in the sentence, not "transmits".
We have changed the word transmits to transmit.
Algorithm 1, line 46: The index "b" is not defined in Algorithm 1. It is probably a beta index.
Yes, the index is beta and we have corrected in the revised manuscript.